# Facial Recognition Method Based on Thin-Film Solar Cells

**Ruei-Tang Chen *** and **Fong-Long Wu**

Department of Electro-Optical Engineering, Southern Taiwan University of Science and Technology, Tainan City 71005, Taiwan; 4a4l0004@stust.edu.tw
* Correspondence: raychen@stust.edu.tw

**Abstract:** In this study, we developed a new facial recognition system using thin-film solar cells as sensors. When the face of a user is illuminated by LED lights on the left and right sides of the system and the reflected light enters the cells at the corresponding positions, differences in facial skin colors and 3D contours lead to different output voltages and currents of the thin-film solar cells. This is the basis of facial feature identification. We found that the accuracy of thin-film-solar-cell-based facial recognition can be improved by precisely controlling changes in LED light intensity. The facial features of six different users were successfully distinguished by this method, thus verifying that thin-film solar cells can be used for green power generation, as well as for facial recognition.

**Keywords:** thin-film solar cell; sensor; LED; facial recognition

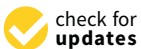



## 1. Introduction

In recent years, facial recognition has become the most widely accepted biometric technology. Presently, it is the only technology among the various biometric techniques with a long-range performance. This recognition technology is realized by capturing facial images with photographic equipment for face model calibration, facial feature extraction, and feature matching. The most complex procedure, namely, feature matching, is performed by analyzing features in 2D images using a series of complicated algorithms, such as Eigenface [1], Fisherface [2], and Local Binary Pattern (LBP) [3,4], to determine whether there is a match.

However, facial recognition is a controversial biometric technology because facial feature matching is based on images captured by cameras across public spaces. Therefore, people may be monitored and photographed for face matching without their knowledge and consent. Such practices violate privacy and human rights [5]. We developed a facial recognition technology that does not rely on photographs or facial recognition algorithms to address this problem. Furthermore, we considered the global trend of green energy development and carbon-emissions reduction [6,7]. Thus, we selected thin-film solar cells as sensing elements to replace traditional camera lenses, thereby introducing a new technique for facial recognition.

In addition to the advantage of high photoelectric conversion efficiency in low light conditions [8,9], thin-film solar cells can be utilized as detectors, as demonstrated in several studies. In 2017, Orgiu et al. [10] developed a perovskite-based oxygen sensor to detect variations in oxygen concentration in a closed environment. The perovskite element in the sensor could increase the current with increasing oxygen concentration. Jong-Min Oh found that perovskite elements could also be fabricated into humidity sensors [11]. The author successfully detected humidity changes through the capacitance variations of the perovskite element. Perovskite solar cells (PSCs) have been employed as refractometric sensors [12]. Similar light sensors also include the DSSC-based optical sensor developed by Rahmadwati et al. [13] and the wide-band photodetector composed of flexible CIGS, which was developed by Qiao et al. [14].

These studies were based on a single thin-film solar cell element, which can be easily customized into small- and medium-sized modules. The application of thin-film solar cells in small- and medium-sized self-powered products has not yet been realized but is anticipated.

The facial recognition system developed in this study uses a 4 × 3 thin-film solar cell array as the sensing element, with an overall sensing area of approximately B5 size. The thin-film solar cell array is self-powering because it absorbs ambient light to generate electricity for storage when the system is not operational. During the operation, the 4 × 3 array receives reflected light after the user's face is illuminated by LEDs. The higher the intensity of the reflected light received, the greater the generated voltage or current. Twelve sets of voltage data were measured and then compared with pre-stored user data. This facial recognition system successfully distinguished six users, indicating the system's suitability for domestic biometric door locks. In such applications, thin-film solar cells can realize their new value, while reducing energy consumption.

## 2. Experimental Design

### 2.1. Fabricating the Facial Recognition System Based on Thin-Film Solar Cells

Twelve SP4.2-37 silicon thin-film solar panels (PowerFilm Inc., Ames, IA, USA) were arranged in a 4 × 3 array (Figure 1). Each panel was 84.0 mm × 36.5 mm and could generate 21 mW of power under 1/4 sun. In the top row, the solar cells from right to left were numbered as SC01 to SC04, respectively. The middle row comprised cells SC05 to SC08, and the bottom row comprised cells SC09 to SC12 (Figure 1). This array served as the sensor for facial recognition and was also part of the system's power source. When the system was not in operation, the array generated electricity to be utilized during operation. Arduino MEGA2560 microcontroller board (https://store-usa.arduino.cc/) (1 December 2021) was used as the system's core for data processing (Figure 2), which reads the voltage generated by each solar cell through 12 analog inputs. The lithium battery stores electricity generated by the solar cells and supplies it to the system and LEDs, distinguishing this system from other biometric technologies requiring external power supplies.

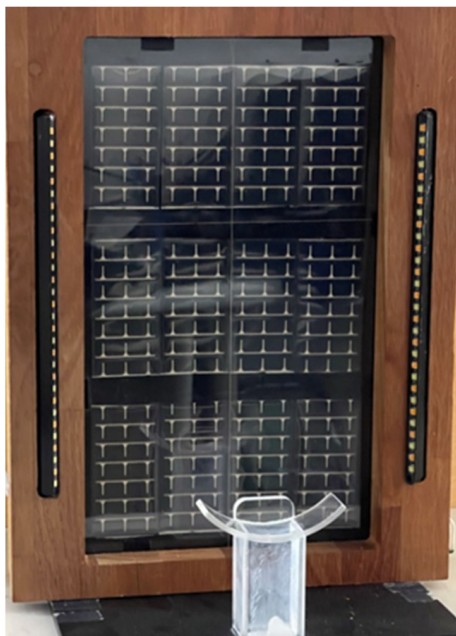

**Figure 1.** Appearance of the facial recognition system.

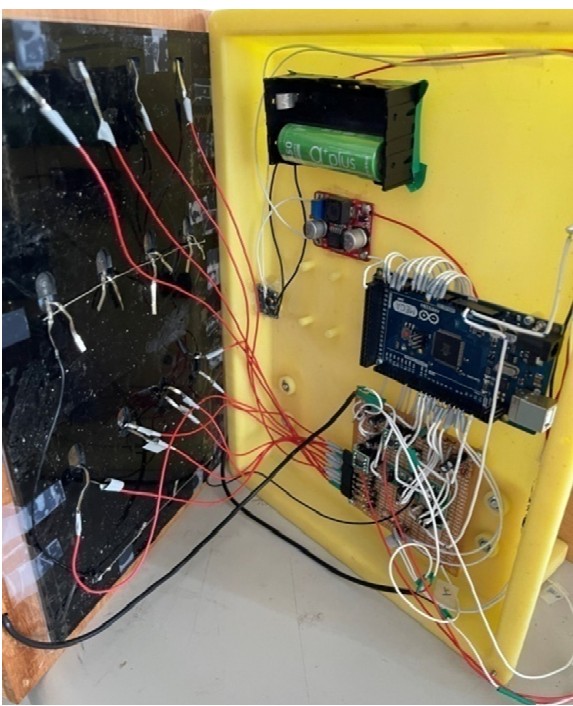

**Figure 2.** Internal configuration of the facial recognition system.

The facial recognition mechanism of this system used two commercially available LED light strips on the left and right sides of the thin-film solar cell array to illuminate the face of the tester at 45°. The Arduino MEGA2560 read the voltage of the twelve thin-film solar cells in the array. The voltage values were affected by the intensity of the reflected light after the LEDs illuminated the face of the user, which served as the basis for facial feature identification. The detection process is demonstrated in Figure 3. We first established a user database and then selected the user's data as a reference. The system could be unlocked if the tester and the reference user were recognized as the same person. When a tester reached the correct physical position for facial recognition, the LEDs were activated for auxiliary illumination. As facial features and skin colors differ between individuals, the intensity of the reflected light entering the thin-film solar cell array after the LED illumination also differ. Accordingly, the voltage values read by the Arduino MEGA2560 differed. The measured voltage values were compared with the documented reference data. If the two sets of values matched, the tester and the reference user were recognized as the same person, and the system was unlocked. Alternatively, if the tester was recognized as a different person, the system could not be unlocked.

*2.2. LED Illumination Control*

The LED lights on the two sides of the facial recognition system were essential because they illuminated the users' faces to identify differences in facial features. Therefore, the stability of the LED light source was one of the most critical considerations in this study, and the quality of the test values was directly related to the stability of the LED light source. Appropriate LED illuminance can emit sufficient light to be reflected and absorbed by the thin-film solar cells. Thus, precisely controlling the stability of the LED illumination was a primary technical issue.

In this study, we used CAT4104 LED driver IC (onsemi, Phoenix, AZ, USA) to precisely control the LED light source of the facial recognition system. CAT4104 is a four-channel LED driver IC that can provide up to 175 mA of current per channel, with the output current controlled by an $R_{Set}$ pin. After assembling the circuit, CAT4104 was used for testing. To evaluate the stability of the LED illumination, the illuminance of the left and right LED light strips was measured in a laboratory environment by using a YFE YF-1065

digital light meter placed 8.5 cm away from the facial recognition system. We repeatedly drove the left and right LED light strips with 131 mA (20 times). The measurement results are listed in Table 1. Although the illuminance of commercially available LED light strips is non-uniform, the left and right LED light strips can still provide stable illumination, indicating that the performance of the CAT4104 LED driver IC on illumination control is satisfactory.

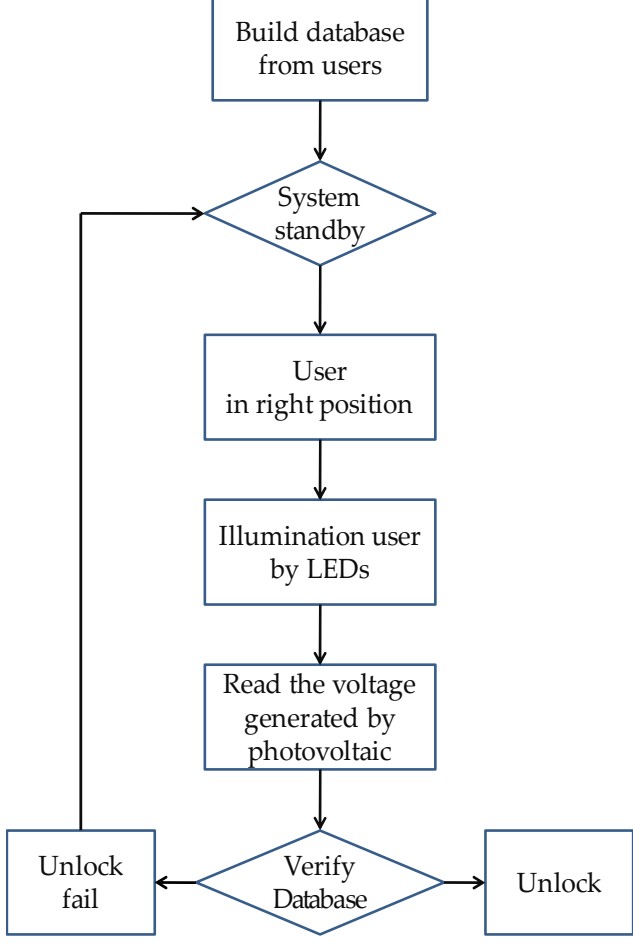

**Figure 3.** Facial recognition flow diagram.

**Table 1.** Illuminance stability of the left and right LED light strips tested 20 times under 131 mA.

|  | Data 1 | Data 2 | Data 3 | Data 4 | Data 5 | Data 6 | Data 7 | Data 8 | Data 9 | Data 10 |
|---|---|---|---|---|---|---|---|---|---|---|
| Left light strip | 658 | 658 | 656 | 657 | 656 | 657 | 657 | 656 | 657 | 656 |
| Right light strip | 694 | 694 | 694 | 694 | 695 | 695 | 696 | 694 | 694 | 693 |
|  | Data 11 | Data 12 | Data 13 | Data 14 | Data 15 | Data 16 | Data 17 | Data 18 | Data 19 | Data 20 |
| Left light strip | 656 | 657 | 656 | 658 | 656 | 657 | 656 | 657 | 658 | 657 |
| Right light strip | 694 | 696 | 696 | 696 | 693 | 694 | 695 | 695 | 694 | 693 |

Unit: Lux.

We measured the spectrum of the white LED light (Figure 4). The spectra in the wavelength range of 400 nm to 500 nm belong to blue light, and those between 500 nm and 600 nm belong to yellow light. Therefore, the white light of the LEDs used in this study was obtained using blue light with yellow fluorescent powder. The spectrogram indicated that the spectra of the left and right LED lights were similar, and their illumination was stable in practical applications. Further experiments with stably controlled LED light sources

revealed that appropriate LED illuminance could help improve the recognition rate of the facial recognition system based on the thin-film solar cells.

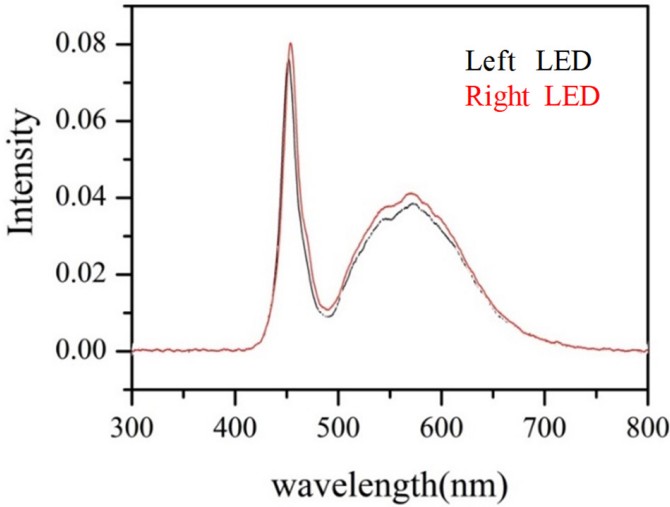

**Figure 4.** White light spectra of the left and right LED light strips.

### 2.3. Facial Recognition Test Parameters

After confirming the test procedure and the reliability of the LED illumination, we drove the LEDs with 131 mA and collected six sets of test data for averaging within 1.5 s of illumination. The data were subsequently compared. The six sets of data are illustrated in Figure 5, and the values are found to be reasonably stable.

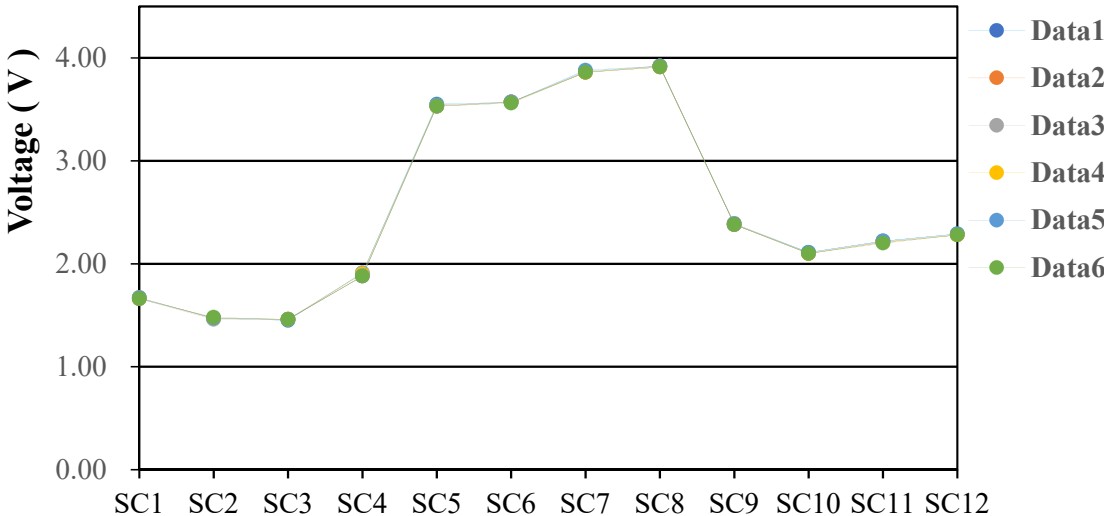

**Figure 5.** Comparisons of six consecutive sets of test data.

## 3. Results and Discussion

After the facial recognition system was established, it was tested with six testers. The photographs of the six testers demonstrated different characteristics in terms of hair length, skin color, and facial contour. The system's ability to identify differences among the acquired data and correlate them with the features in the photographs is critical to its success as a facial recognition system. For a detailed investigation, we also conducted 3D contour scanning for the testers using an iPhone 12 Pro and Bellus3D FaceApp (Figures 6–11).

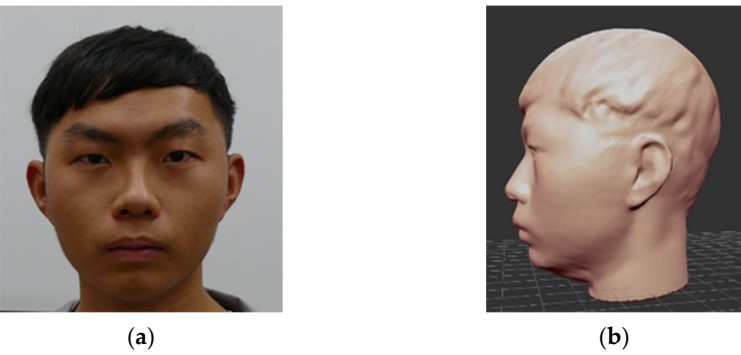

**Figure 6.** Tester 1 (**a**) and his side-view 3D scan image (**b**).

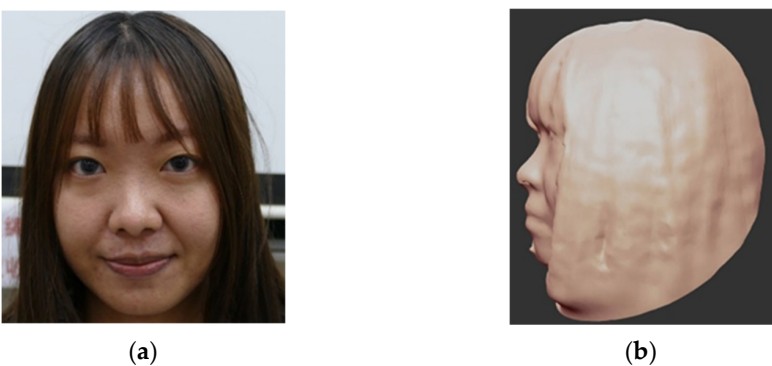

**Figure 7.** Tester 2 (**a**) and her side-view 3D scan image (**b**).

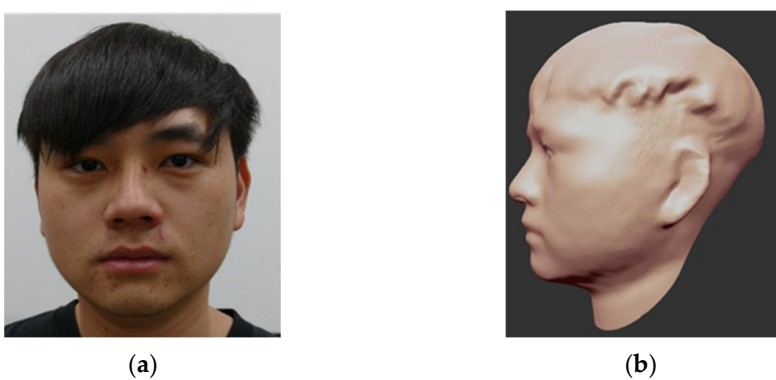

**Figure 8.** (**a**) Tester 3 and (**b**) his side-view 3D scan image.

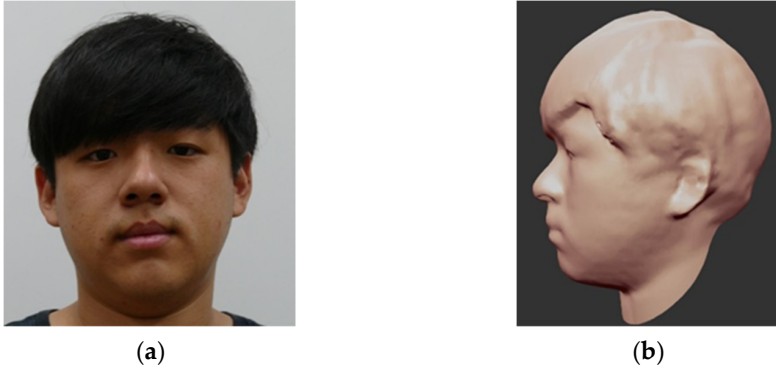

**Figure 9.** (**a**) Tester 4 and (**b**) his side-view 3D scan image.

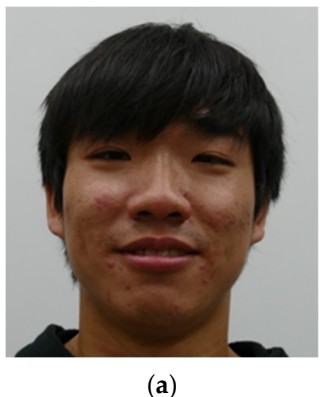 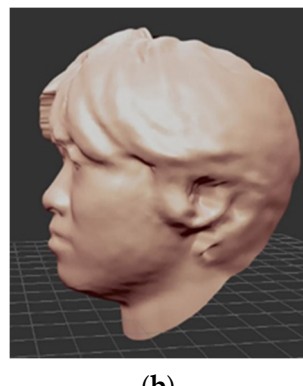

(**a**)　　　　　　　　　　　　　　　(**b**)

**Figure 10.** (**a**) Tester 5 and (**b**) his side-view 3D scan image.

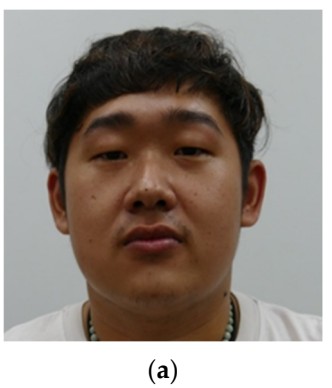 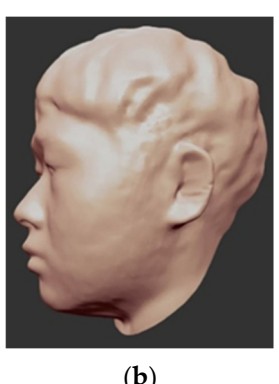

(**a**)　　　　　　　　　　　　　　　(**b**)

**Figure 11.** (**a**) Tester 6 and (**b**) his side-view 3D scan image.

To avoid plotting the data for all testers on the same graph, we chose Tester 1 as a reference to separately compare his data with the others. The facial areas corresponding to the solar cells are demonstrated in Figure 12. SC01–SC04 correspond to the forehead, SC05–SC08 correspond to the eyes, cheeks, and nose, and SC9–SC12 correspond to the mouth and chin.

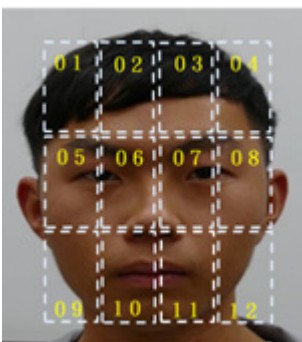

**Figure 12.** Correspondence between solar cells and different facial areas.

Figure 13 illustrates that the Tester 2 voltage values of SC01, SC04, SC05, and SC08 are lower than those of Tester 1, which is affected by hair. The facial areas of Tester 2 corresponding to SC05 and SC08 (the cheeks) are covered by hair, resulting in lower voltages values. For Tester 1, the value of SC04 is higher than that of SC01 because of a slanted fringe. The exposed forehead area corresponding to SC04 is larger, so the reflected light of LED illumination entering the thin-film solar cell has higher intensity, and the voltage value is higher.

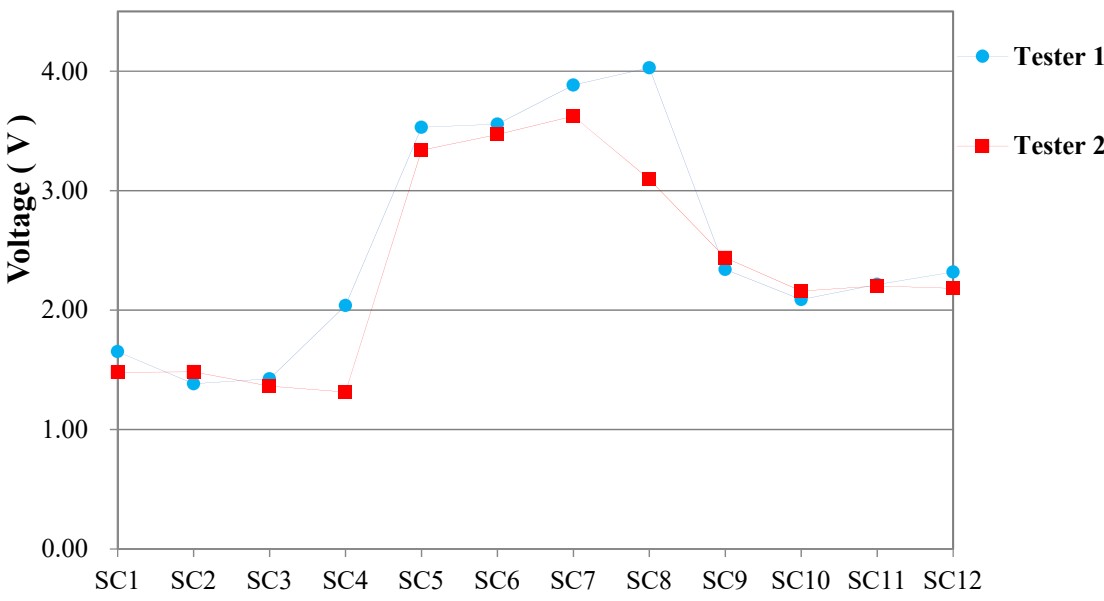

**Figure 13.** Differences between the Tester 1 and Tester 2 data.

In Figure 14, the Tester 3 voltage values of SC01–SC04 are lower than those of Tester 1 because Tester 3 has more hair around the facial frame. Moreover, the skin color of Tester 3 is lighter, so the Tester 3 voltage values of SC05–SC12 are generally higher than those of Tester 1.

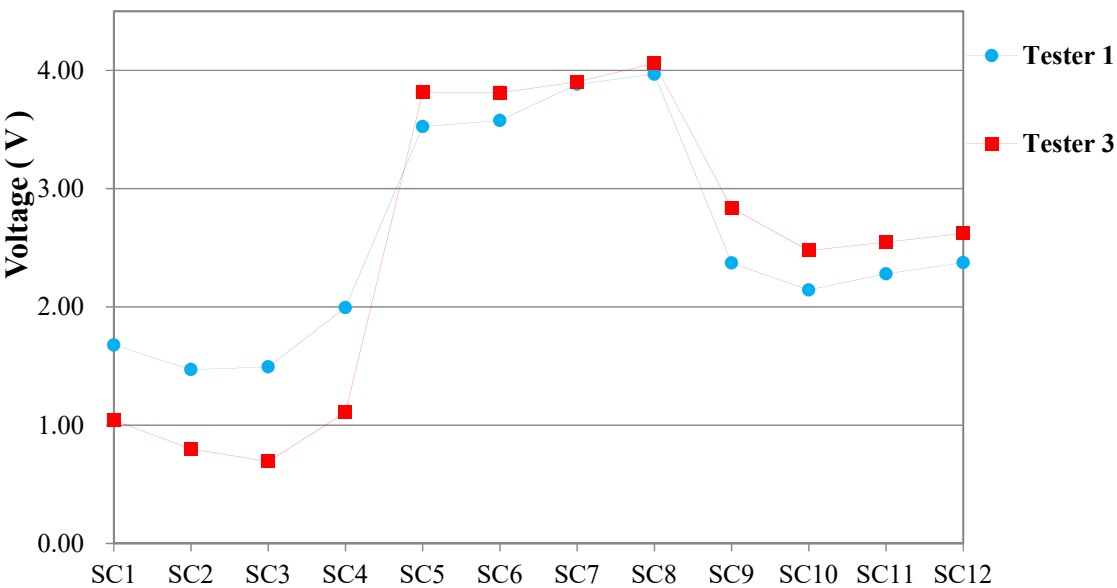

**Figure 14.** Differences between the Tester 1 and Tester 3 data.

Tester 4 has more hair around the facial frame than Tester 1, and therefore, the Tester 4 voltage values of SC01–SC04 are lower than those of Tester 1 (Figure 15). The SC08 data may be affected by the hair below the ears (sideburns). Tester 1 has fewer sideburns, and there is less light absorbed by the sideburns, resulting in a higher voltage value than that of Tester 4. For the right chin area corresponding to SC09, Tester 4 has a slightly more rounded chin, and the Tester 4 SC09 voltage is slightly higher than that of Tester 1.

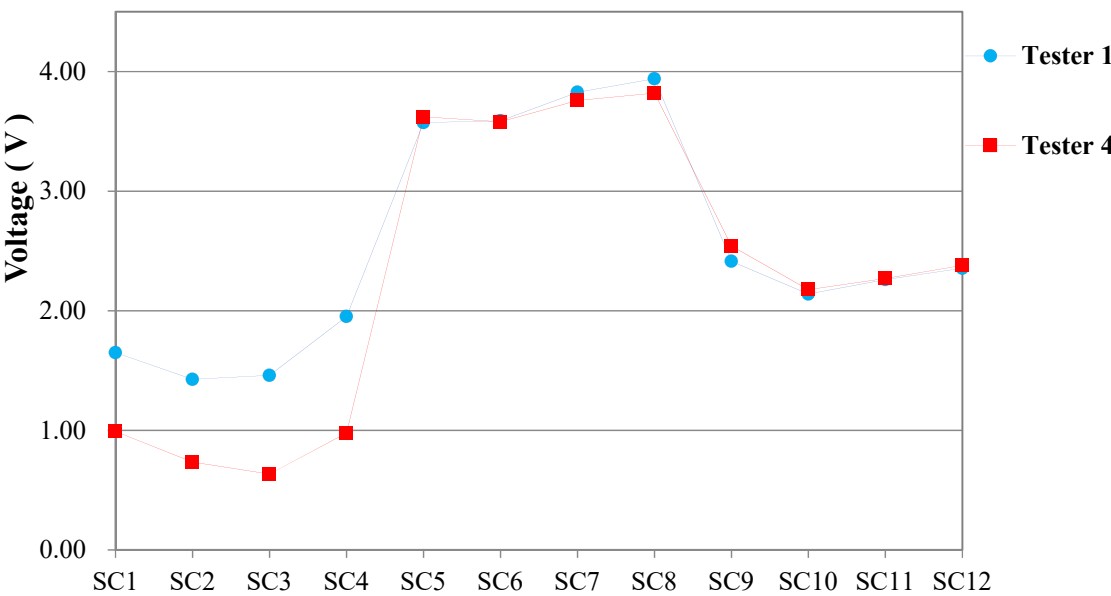

**Figure 15.** Differences between the Tester 1 and Tester 4 data.

The differences in the values of SC01–SC04 corresponding to the forehead are significant because of the differences in the testers' hairstyles (Figure 16). For SC06 and SC07, we can observe from the 3D scan images that the facial areas on both sides of the nose of Tester 5 are raised, whereas the same areas of Tester 1 are flat. A flat surface favors light reflection, but a raised surface generates diffused reflection, and hence, reduced intensity of the light entering the sensing element. Therefore, the Tester 5 SC06 and SC07 data are lower than those of Tester 1.

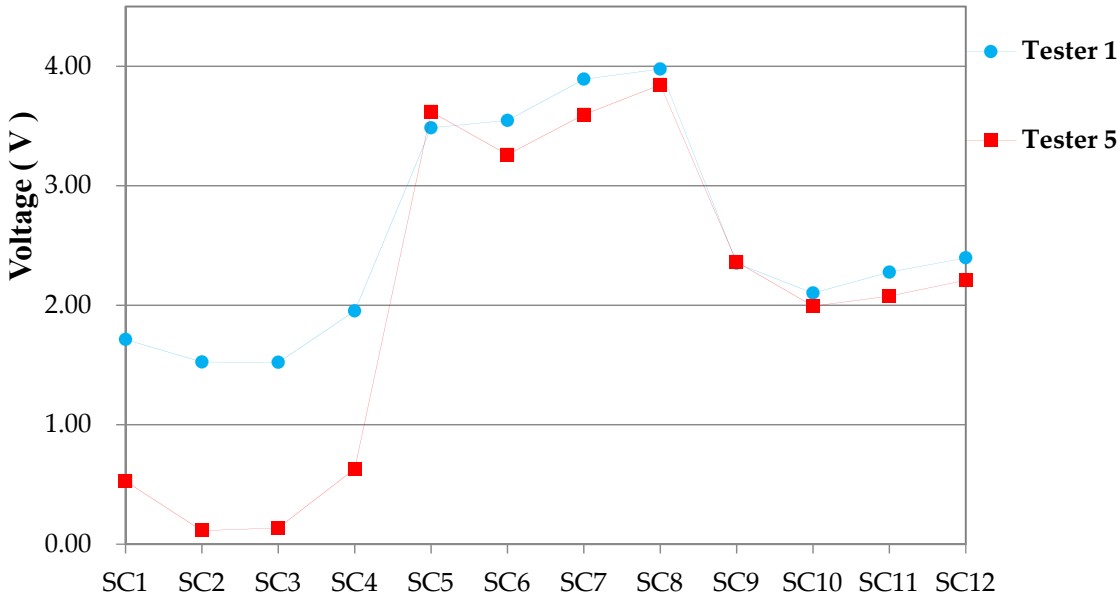

**Figure 16.** Differences between the Tester 1 and Tester 5 data.

The photographs indicate that the face of Tester 6 is rounder. Therefore, the of SC05 and SC08 values for Tester 6 are higher than those of Tester 1 (Figure 17). In addition, the skin color of Tester 6 is lighter, so the overall data values of Tester 6 are also higher. The 3D model of Tester 6 shows that facial areas on both sides of the nose are slightly raised, indicated by the shallows. The reflected diffusion decreases the intensity of the light

entering the sensing element, so the Tester 6 voltage values of SC06 and SC07 are lower than those of Tester 1.

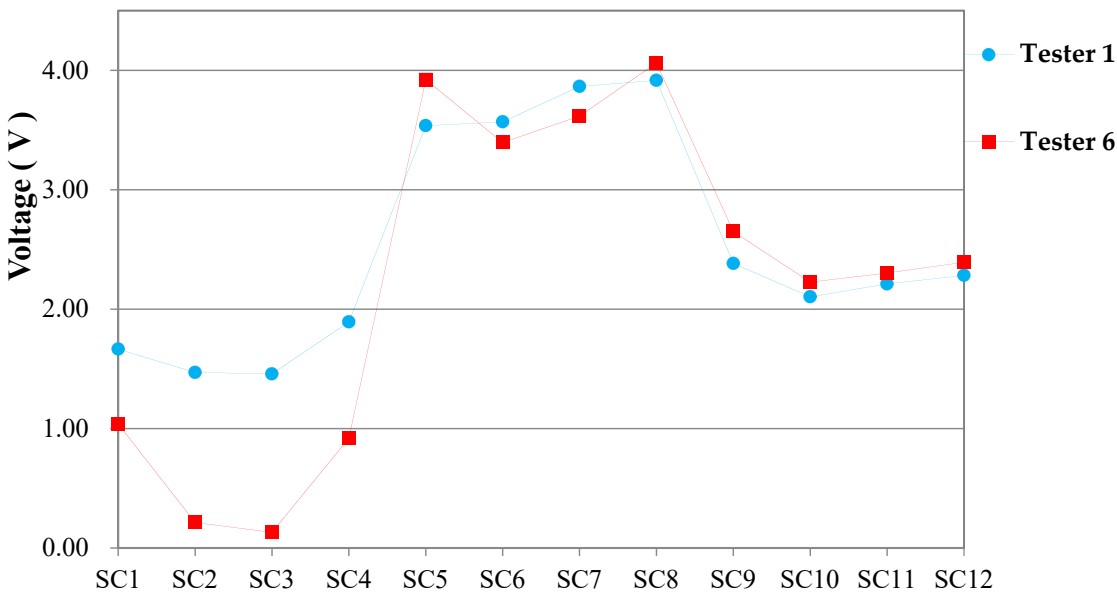

**Figure 17.** Differences between the Tester 1 and Tester 6 data.

Although we successfully identified six different testers, the SC05–SC08 values of several testers were similar, which may be limited by small voltage changes of the solar cell in high-light environment. Ideally, current variations in high-light environments should also be detected; however, a micro-current detection element suitable for the Arduino MEGA2560 has not been found. If we can divide the voltage records of each solar cell in our present system into eight intervals, then the array with 12 solar cells will generate 8 to the 12th power of possibilities ($8^{12}$).

## 4. Conclusions

With the technological development, traditional facial recognition techniques based on high-resolution CCD and increasingly complex algorithms can successfully identify multiple faces simultaneously from a considerable distance. This achievement has exceeded the general public's expectations for biometric applications and has made people feel that they are constantly under surveillance. In this study, we established a close-range non-photographic facial recognition system based on feature comparison using a thin-film solar cell array. This system detects light reflected off the user's face as it enters the thin-film solar cell array, the intensity of which is affected by the user's skin color and 3D facial contours. Then, it compares the voltage output of the solar cell array with the pre-stored database to determine whether the user can unlock the system. In this study, we succeeded in distinguishing six users, demonstrating that thin-film solar cells can participate in green power generation and be employed in facial recognition systems.

**Author Contributions:** Conceptualization, R.-T.C.; methodology, R.-T.C. and F.-L.W.; software, F.-L.W.; validation, R.-T.C. and F.-L.W.; formal analysis, R.-T.C. and F.-L.W.; investigation, R.-T.C. and F.-L.W.; resources, R.-T.C.; data curation, F.-L.W.; writing—original draft preparation, R.-T.C. and F.-L.W.; writing—review and editing, R.-T.C.; visualization, R.-T.C. and F.-L.W.; supervision, R.-T.C.; project administration, R.-T.C.; funding acquisition, R.-T.C. All authors have read and agreed to the published version of the manuscript.

**Funding:** This research was funded by Southern Taiwan University of Science and Technology.

**Institutional Review Board Statement:** Not applicable.

**Informed Consent Statement:** Not applicable.

**Data Availability Statement:** Not applicable.

**Acknowledgments:** This work was partly supported by the Southern Taiwan University of Science and Technology.

**Conflicts of Interest:** The authors declare no conflict of interest.

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
