# Peer review of "Facial Recognition Method Based on Thin-Film Solar Cells"

_applsci, doi:10.3390/app12031157_

Round 1

Reviewer 1 Report

An extensive editing of English language and style is required.

In the manuscript titled "The Naturally Face Recognition Method Base on Green Technology" the authors described the synthesis of carbohydrate triazole derivatives by using copper nanoparticles as catalysts. A new facial recognition system using thin film solar cells as sensors is used. When the users’ faces are illuminated by LED lights on the left and right sides of the system, the differences in their facial skin colors and 3D contours will lead to different output voltages and currents of the thin film solar cells as the reflected light enters the cells at the corresponding positions, the subject is not covered by any of the journal's areas.

The use of solar cells for facial recognition cannot be considered as green technology.

The authors have to improve the discussion about their results.  They should compare their results to the related literature.

Figure 4: the quality has to be improved.

The experimental part does not include any preparation, it is about assembling ready materials.

Author Response

Response to Reviewer 1 Comments

In the manuscript titled "The Naturally Face Recognition Method Base on Green Technology" the authors described the synthesis of carbohydrate triazole derivatives by using copper nanoparticles as catalysts. A new facial recognition system using thin film solar cells as sensors is used. When the users’ faces are illuminated by LED lights on the left and right sides of the system, the differences in their facial skin colors and 3D contours will lead to different output voltages and currents of the thin film solar cells as the reflected light enters the cells at the corresponding positions, the subject is not covered by any of the journal's areas.

The use of solar cells for facial recognition cannot be considered as green technology.

Point 1: The use of solar cells for facial recognition cannot be considered as green technology.

Response 1: The system uses thin-film solar cells as sensors in a new type face recognition device. When the system is not operating, the thin-film solar cells can receive ambient light to generate electricity, which is stored in the built-in battery in the system.

Point 2: The authors have to improve the discussion about their results.  They should compare their results to the related literature.

Response 2: This system is the first face-recognition device that uses thin-film solar cells as sensors. Thus, there are no corresponding literature results for comparison.

Point 3: Figure 4: the quality has to be improved.

Response 3: The quality of Figure 4 has been improved.

Point 4: The experimental part does not include any preparation, it is about assembling ready materials.

Response 4: Thank you for this suggestion. We have added a brief description of the system assembly.

Reviewer 2 Report

The authors study the new facial recognition system using thin-film solar cells. I recommend its publication after the authors address the following comments:

1- The authors should compare their results with previous studies and highlight the novelty of this work compared to previous studies.

2- Which type of LED did you use? I mean that the wavelength of the LED is not important for your measurements?

3- Which type of solar cells did you use? 

4- In figure 5, you showed the values are reasonably stable? Can you explain this more clearly?

5- Based on plotting the voltage as a function of the Sc1, and SC2, etc., how you can find the difference between the two tasters? I mean that you show this difference because of the hairstyle of skin color. Without having the initial information about the taster we can plot and find the facial shape of someone?

Author Response

Response to Reviewer 2 Comments

The authors study the new facial recognition system using thin-film solar cells. I recommend its publication after the authors address the following comments:

Point 1: The authors should compare their results with previous studies and highlight the novelty of this work compared to previous studies.

Response 1: Thank you for this suggestion This system is the first face-recognition device that uses thin-film solar cells as sensors. Thus, there are no corresponding literature results for comparison. In contrast to the traditional facial recognition system, which recognizes a complete photo through a complex recognition program, our novel system directly recognizes the user based on the light reflected by the facial skin color and 3D outline.

Point 2: Which type of LED did you use? I mean that the wavelength of the LED is not important for your measurements?

Response 2: We use commercially available LED strips in this system. The color of light theoretically affects the accuracy of identification. Preliminary data indicate that the reflectivity of yellow LEDs is higher in people with yellow-hued skin, that  should improve the accuracy of identification.

Point 3: Which type of solar cells did you use?

Response 3: We have used liquid DSSCs and solid-state DSSCs and CIGS cells. The proposed system contains silicon thin-film solar cells.

Point 4: In figure 5, you showed the values are reasonably stable? Can you explain this more clearly?

Response 4: As stated by the reviewer, Figure 5 depicts the stability of the data during measurement. The data were stable and consistent across repeated tests performed by the same tester over a short period of time. However, variance in measurements can occur owing to changes in hair and beard styles  after long durations.

Point 5: Based on plotting the voltage as a function of the Sc1, and SC2, etc., how you can find the difference between the two tasters? I mean that you show this difference because of the hairstyle of skin color. Without having the initial information about the taster we can plot and find the facial shape of someone?

Response 5: An important feature of our system is that it cannot draw the face shape through voltage data  The data of 12 measurement points cannot construct the real face of the user; hence, there is no possibility that traditional facial recognition may violate human rights.

Reviewer 3 Report

In this work, the authors presented a facial recognition system using thin film solar cells as sensors. It detects the reflected light off the user’s face as it enters the thin film solar cell array, whose intensity is affected by the user’s skin color and 3D face contours. It compares the voltage output of the solar cell array with the pre-stored database to determine whether the user can unlock the system. The method is also of scientific interest and technological presence. Thus, I suggest its publication in Applied Sciences after addressing the following issues.

  1. Compared to the traditional CCD facial recognition technique, what is biggest advantage of this TFPV-based facial recognition system?
  2. The authors are suggested to provide a specific evaluation index, e.g., resolution, accuracy to describe this thin film solar cell facial recognition system, especially for plenty of testers.
  3. In this work, LED light intensity induced output voltage variations are tested and compared, the authors have any idea to add the information of current variations in this system?
  4. The silicon thin film solar cells are used in this work, any consideration of the traditional thin film solar cells of CdTe, CIGS etc.
  5. Some low-cost and efficient TFPVs are also suggested to expect in this facial recognition system, e.g., Sb2Se3 in 10.1016/j.nanoen.2020.104806, CZTS in 10.1038/s41467-021-23343-1, considering their superior weak-light response and high voltage output.

Author Response

Response to Reviewer 3 Comments

In this work, the authors presented a facial recognition system using thin film solar cells as sensors. It detects the reflected light off the user’s face as it enters the thin film solar cell array, whose intensity is affected by the user’s skin color and 3D face contours. It compares the voltage output of the solar cell array with the pre-stored database to determine whether the user can unlock the system. The method is also of scientific interest and technological presence. Thus, I suggest its publication in Applied Sciences after addressing the following issues.

Point 1: Compared to the traditional CCD facial recognition technique, what is biggest advantage of this TFPV-based facial recognition system?

Response 1: There are two unique advantages of this TFPV-based facial recognition system compared with the traditional CCD face-recognition technology. First, thin-film solar cells simultaneously act as sensors and generate electricity. Second, our system cannot lead to violations of human rights.

Point 2: The authors are suggested to provide a specific evaluation index, e.g., resolution, accuracy to describe this thin film solar cell facial recognition system, especially for plenty of testers.

Response 2: In our system, identification relies on users’ skin color and 3D contours. Among a large number of testers, accidental unlocking may occur because of similar skin colors and 3D contours. Just like a password lock, a 4-digit password lock will have a one-in-ten-thousand chance of being cracked. The probability of unlocking by mistake can be reduced by increasing the number of thin-film solar cell arrays. The data accuracy of the same tester is extremely high when the device is tested over a short period of time, and unlocking can be successful in repeated tests. However, over long durations, the success rate of unlocking will significantly decrease if the length of the hair changes.

Point 3: In this work, LED light intensity induced output voltage variations are tested and compared, the authors have any idea to add the information of current variations in this system?

Response 3: Thank you for this question. We shall further study variations in this system after obtaining complete data The initial testing results indicate that highly reflective areas (such as cheeks) or low reflective areas (such as hair) influence LED light intensity. Identification is improved if the LED light intensity in high reflectivity areas is lowered and the intensity is increased in low reflective areas. These adjustments depend on the varieties of thin-film solar cells.

Point 4: The silicon thin film solar cells are used in this work, any consideration of the traditional thin film solar cells of CdTe, CIGS etc.

Response 4: Liquid and solid-state DSCs or CIGS cells were used in this research. CdTe is expected to be used . However, this manuscript describes results obtained using silicon thin-film solar cells.

Point 5: Some low-cost and efficient TFPVs are also suggested to expect in this facial recognition system, e.g., Sb2Se3 in 10.1016/j.nanoen.2020.104806, CZTS in 10.1038/s41467-021-23343-1, considering their superior weak-light response and high voltage output.

Response 5: As mentioned by the reviewer, solar cells with high voltage output or high photoelectric conversion efficiency in weak-light can be used in this system. In our previous research, the homemade DSCs and CIGS cells were also applied in the system; however, the cells were unstable.

Round 2

Reviewer 2 Report

your answers to comments are good and I think it paper can be published in its current format in the journal.

Reviewer 3 Report

The responses seems acceptable, though it can be further improved.